# Influence of Breast Cancer Extracellular Vesicles on Immune Cell Activation: A Pilot Study

**DOI:** 10.3390/biology12121531

**Published:** 2023-12-15

**Authors:** Jessie Santoro, Barbara Carrese, Maria Sara Peluso, Luigi Coppola, Massimiliano D’Aiuto, Gennaro Mossetti, Marco Salvatore, Giovanni Smaldone

**Affiliations:** 1IRCCS SYNLAB SDN, Via E. Gianturco, 80143 Naples, Italy; jessie.santoro@synlab.it (J.S.); maria.peluso@synlab.it (M.S.P.); direzionescientifica.irccssdn@synlab.it (M.S.); giovanni.smaldone@synlab.it (G.S.); 2Clinica Villa Fiorita, Via Filippo Saporito, 24, 81031 Aversa, Italy; info@daiuto.it; 3Pathological Anatomy Service, Casa di Cura Maria Rosaria, Via Colle San Bartolomeo, 50, 80045 Pompei, Italy; drmossetti@libero.it

**Keywords:** breast cancer, tumor microenvironment, anti-cancer immunity, extracellular vesicles, spheroids, personalized medicine, innovative diagnostic approach

## Abstract

**Simple Summary:**

The tumor microenvironment (TME) is a complex system composed of different cellular components that contribute to tumor growth and sustenance. The ability of the tumor to interact with the immune system through extracellular vesicles (EVs) represents a fundamental challenge for the scientific community to shed light on the intricate system that the tumor develops to escape the body’s control. In our study, we investigate a possible specific immune cell activation towards releasing EVs and whether such activation could differ between different breast cancer subtypes or cellular 2D/3D model growth. Our data, although preliminary, aim to shed light on the effect of TME on the anti-tumor immune component to further develop personalized approaches in cancer diagnosis and treatment.

**Abstract:**

Breast cancer is the leading cause of cancer-related death in women worldwide. It is well known that breast cancer shows significant alterations in the tumor microenvironment (TME), which is composed of a variety of immune cells, including natural killer (NK) cells, that have a key role in tumor development or anti-tumor responses in breast cancer patients. Luminal B (BT474) and triple-negative breast cancer (HS578T) cell lines were cultured in 2D and 3D model systems. PMBCs from healthy donors were isolated and treated with extracellular vesicles (EVs) from monolayer and spheroids of BT474 and HS578T and analyzed using cytofluorimetric approaches. We observed that EVs can alter the activation and presence of CD335+/CD11b+ NK cells. EVs derived from BT474 and HS578T cells trigger the activation and, simultaneously, a reduction in the percentage of CD335+/CD11b+ NK cells. In addition, EVs derived from BT474 also significantly reduce CD39+ T-regulatory (T-reg) cells. Our preliminary data suggest that using EVs to treat tumors could potentially alter components of the immune system, which causes hyperactivation of specific cell types and can lead to aggressive growth. These data will guide the designing of new personalized diagnostic approaches based on in-depth study of the TME.

## 1. Introduction

Breast cancer is the most frequently diagnosed cancer among women and the leading cause of cancer-related death in women worldwide [1]. This neoplasm shows cellular and molecular heterogeneity associated with receptor status (ER, PR, HER2) [2], associated with the presence of different clones of tumor cells and the composition of the tumor stroma. Recent studies demonstrated that tumor epithelial cells can only thrive in an altered microenvironment in which several components are involved, including immune cells [3]. Breast tumors are not considered immunogenic due to their limited response to immunotherapy. Nevertheless, the role of the breast immune microenvironment has recently been re-evaluated [4]. The tumor microenvironment refers to the complex and dynamic ecosystem that surrounds and interacts with cancer cells within a tumor; it consists of a variety of cell types, molecules, and structures that influence the growth, progression, and behavior of the tumor [5]. Understanding the tumor microenvironment is essential in cancer research and therapy because it plays a significant role in tumor development, immune response, aggressiveness, and resistance to treatment [6]. Nowadays, NK cells are attracting increasing interest in the scientific community. The role of NK cells in different tumor types is controversial and further insights are needed to better understand their function [7]. The NK cells are classified into two subsets based on the expression level of CD56 and CD16 [8]. A new classification of NK cells is based primarily on the expression of CD56 and CD11b, an integrin family member used to identify the mature peripheral component of NK cells [9]. Other maturation markers [10] such as CD335 and activation markers such as HLA-DR [11] are used to investigate the role of these cells in breast cancer patients. It is important to underline that other actors are involved in the regulation of the activity of NK cells in the tumor microenvironment. Among them, T-reg cells may act as key mediators of peripheral tolerance by specifically preventing immune responses. Elevated T-reg presence in breast cancer biopsies is linked to an invasive phenotype and decreases recurrence-free and overall survival [12]. To fully understand the breast tumor microenvironment, it is crucial to study the mechanisms underlying the interaction between tumor and immune cells. In this scenario, extracellular vesicles (EVs) play a key role [13]. EVs are membrane-contained vesicles released from all types of cells in both physiological and pathological conditions. Moreover, EVs contain nucleic acids and proteins that are released into the extracellular environment [14]. They mediate cell-to-cell communication and modulate their function; however, EVs released by cancer cells may play an important role in shaping the tumor immune microenvironment, which is still understudied [15]. In the tumor environment, EVs can have different functions [16]. For example, tumor-derived EVs can participate in the formation of the premetastatic niche since they carry proteins or miRNAs associated with cell-tight junctions [17]. These particles can promote tumor invasion and metastasis through the modification of epithelial cells and induction of epithelial–mesenchymal transition (EMT) [18,19,20].

Although monolayer cell cultures represent one of the most frequently used methods to separate EVs due to their low cost, easiness, and reproducibility, unfortunately, 2D models do not mimic the characteristics and architecture of in vivo tumors. Nowadays, to overcome these issues, spheroids, cell aggregates formed when cells are maintained on a substrate that does not allow their adhesion, are widely studied. Several spheroid formation methods were developed, including hanging drop, scaffolds and hydrogels, to test drugs and nanoparticles as well as to model disease [21] because they can simulate cell-to-cell and cell-to-extracellular matrix (ECM) contacts, cellular layered assembling, hypoxia and gradients of nutrient, oxygen, and pH observed in vivo [22]. Furthermore, 3D models show different physical properties, gene expression, and chemosensitivity compared to monolayer culture.

To study the role of the immune system in response to the breast cancer microenvironment in vitro, we used both monolayer and 3D cell models of luminal B (BT474) and triple-negative (HS578T) breast cancer cell lines.

Luminal B breast cancer has lower expression of hormone receptors, while its histologic grade is higher compared to luminal A. It is associated with a worse prognosis and less responsiveness to chemotherapy [23]. Likewise, it is known that triple-negative breast cancer is associated with high invasiveness, tumorigenesis, and poor prognosis. Therefore, personalized approaches for triple-negative breast cancer (TNBC) patients are still lacking [24]. Specifically, a culture medium was used for EV separation from these breast cancer cell subtypes to stimulate peripheral blood mononuclear cells (PBMCs) obtained from healthy women.

The aim of our study was to evaluate the in vitro effects of EVs from two examples of breast cancer subtypes (Luminal B and TNBC) on PBMC from healthy donors. Using a monolayer or spheroid cell growth model, the aim was to determine which model system was best to study the effects of EVs on the tumor microenvironment to fully elucidate the role that tumor-derived EVs may have on the anti-cancer immune response.

## 2. Materials and Methods

### 2.1. Cell Culture and Spheroid Formation

Human breast carcinoma cell lines (HS578T and BT474) were obtained from IRCCS Synlab SDN Biobank [25] and grown in DMEM supplemented with 10% heat-inactivated Fetal Bovine Serum (FBS) (GIBCO), 100 U/mL penicillin, 100 mg/mL streptomycin, and 1% L-glutamine. All cell lines were grown at 37 °C in a 5% CO_2_ atmosphere. HS578T and BT474 spheroids were obtained by seeding cells at a density of 10,000 cells/50 µL in 96-well round (U)-bottom plate with a low-attachment surface (Cat: #7007, Corning, NJ, USA). After 72 h, 50 µL of fresh medium was added to spheroids. The size of each spheroid was observed by confocal microscopy.

### 2.2. EV Separation from 2D and 3D Cell Cultures

Prior to EV separation, BT474 and HS578T 2D and spheroids were seeded at a density of 40,000 cells/2 mL in 6-well plates with EXO-FREE medium for 48 h. Afterwards, EVs were separated following a previous publication [26]. In brief, the medium from 2D and 3D culture was centrifuged at 300× g for 10 min to remove cells, then the supernatant was centrifuged at 2000× g for 10 min to remove dead cells. The remaining cell debris was removed using ultracentrifugation at 10,000× g for 30 min at 4 °C with an OPTIMA MAX-XP ultracentrifuge (Cat: # 393315, Beckman Coulter, Brea, CA, USA). The cleared conditioned medium was ultracentrifuged at 100,000× *g* for 70 min at 4 °C for pelleting EVs. Finally, EV pellets were washed using 0.22 µm-filtered PBS at 100,000× *g* for 1 h at 70 min. EVs were resuspended in 100 µL of 0.22 µm-filtered PBS and stored at −80 °C for further analysis.

### 2.3. Particle Concentration and Size Using Nanoparticle Tracking Analysis (NTA)

The particle concentration and size of BT474 and HS578T 2D and 3D EVs were analyzed using NTA (NanoSight NS300, Malvern Instruments Ltd., Malvern, UK). NTA exploits Brownian motion and light scattering to quantify the particle size and concentration of EVs. EV samples were injected into the NTA system under constant flow conditions (flow rate = 50). Five 60 s videos were recorded and were analyzed using NTA 3.2 software. Three replicates of each sample were analyzed independently using NTA.

### 2.4. Immunoblotting Analysis of EVs

EVs were characterized using immunoblotting analysis. Specifically, EVs were lysed using JS lysis buffer (HEPES 50 mM; NaCl 150 mM; Glycerol 1%; Triton (×100) 1%; MgCl_2_ 1.5 mM; EGTA 5 mM; H_2_O). Extracted EVs protein (30 μg) was resolved on 10% gel using electrophoresis at 120 V, and proteins were transferred using a Trans-Blot Turbo System (Bio-Rad Laboratories, Cat. 690BR024275, Hercules, CA, USA). Filters were blocked with 5% milk in TBST containing 0.1% Tween-20 for 1 h and incubated overnight at 4 °C with primary antibodies anti-TSG101, (1:1000; Cat. ab30871, Abcam, Waltham, MA, USA), anti-CD63 (1:1000; Cat. ab68418, Abcam, Waltham, MA, USA) and anti-calnexin (1:1000; Cat. ab133615, Abcam, Waltham, MA, USA). The secondary antibodies used were Goat Anti-Rabbit IgG (1:2000; Cat. 4030-05, SouthernBiotech, Homewood, AL, USA). Imaging was performed using an automated ChemiDoc™ MP Imaging System (Bio-Rad Laboratories, Cat. 12003154, Hercules, CA, USA) and Clarity Max™ Western ECL Substrate (Cat. 1705062, Bio-Rad Laboratories, Hercules, CA, USA). BT474 and HS578T monolayer cell lysates (CL) were used as a positive control.

### 2.5. PBMC Isolation from Whole Blood

PBMCs were obtained from five healthy women volunteers from a cohort of medical doctors, nurses, technicians, biologists, and nonmedical personnel working in the IRCCS SYNLAB SDN institute enrolled for this study (CE of IRCCS PASCALE Naples, Italy; reference number 4/21, 2021). PBMCs were recovered from venous blood using density gradient centrifugation (Cat. # Pancoll) (https://www.frontiersin.org/articles/10.3389/fimmu.2022.896255/full (accessed on 6 July 2022)). Briefly, whole blood collected in an EDTA vacutainer was diluted in 5 mL PBS, layered on 3 mL Pancoll and centrifuged at 1200× *g* for 10 min at 4 °C. Afterwards, PBMCs were collected and mixed in a new falcon tube with 5 mL of 2% FBS in PBS and centrifuged at 1800 rpm for 5 min at 4 °C. Finally, the pellet was resuspended in 5 mL of 2% FBS-PBS and cells were analyzed for vitality prior to use for further experiments.

### 2.6. PBMCs Treatment with Breast Cancer EVs

Fresh PBMC samples obtained as described above were treated with BT474, HS578T EVs from 2D and 3D culture. Specifically, 4 × 10^5^ PBMCs were pelleted and resuspended with 1 × 10^9^ EVs from 2D and 3D culture of both cell lines in 2% FBS-PBS and incubated for 24 h at 37 °C in a 5% CO_2_. A control was also carried out, where PBMCs were incubated only with 2% FBS-PBS in the same condition.

### 2.7. Flow Cytometry Analyses

After 24 h, all patients’ PBMCs treated as described before were analyzed using CytoFLEX (Beckman Coulter, Brea, CA, USA). Specifically, PBMCs were stained for surface marker detection using the following antibody mixture: CD8-FITC, CD4-PE, CD3-ECD, HLA-DR-PC5, CD335-PC7, CD36-APC, CD56-APC700, CD14-APC750, CD11b-PB, and CD45-KO. Each antibody was prepared in a 1:10 dilution and mixed throughout. All the antibodies used were purchased from Beckman Coulter. Furthermore, for the evaluation of T-reg activity, we used a Duraclone IM T-reg kit (Cat. # B53346, Beckman Coulter). As specified before, all PBMC samples were treated under the same conditions (10^9^ EVs from 2D and 3D culture of both cell lines) and processed following the manufacturer’s instruction. The Kaluza 2.1 software (Beckman Coulter, USA) was used for immune cell characterization.

### 2.8. Statistical Analysis

All the experiments detailed in the study were performed a minimum of 3 separate times. The results of the assays and EV collections are expressed as mean ± SD of three independent experiments. Statistical analysis was performed, and figures were drawn, using GraphPad Prism Version 9. Ordinary one-way ANOVA was used to assess the comparison of EVs separated by different approaches. *p*-values < 0.05 were considered statistically significant. We have submitted all relevant data of our experiments to the EV-TRACK knowledgebase (EV-TRACK ID: EV230984) [27].

## 3. Results

### 3.1. Characterization of EVs Separated from BT474 and HS578T from 2D and 3D Cell Culture

The particle concentration and size in BT474 and HS578T samples were analyzed using NTA, as reported in Table 1. Overall, the numbers of particles in both cell lines either in 2D and 3D conditions were similar (approximately 10^9^ particles/mL) with an average particle size of 170 nm (Figure 1A). Furthermore, we observed a heterogeneous population in EVs separated from the 3D BT474 and HS578T; for the 2D cell condition, a single peak was detected around 140 nm (BT474) and 110 nm (HS578T), as shown in Figure 1A.

### 3.2. Validation of EV Markers through Immunoblotting in 2D and 3D Breast Cancer EVs

To confirm the presence of EVs separated from BT474 and HS578T, we performed immunoblotting analysis using EV markers and following MISEV2018 guidelines [28] (Figure 1B and Appendix A). Overall, we observed that CD63, an EV-associated protein, was expressed in all EV-samples as well as in the cell lysate used as a control. Furthermore, an internal EV marker (TSG101, MISEV guidelines) was detected in all EV samples and in the lysate control. We also used a non-EV-associated marker (as recommended by MISEV guidelines), calnexin, which was not detected in any of the samples analyzed, but only in the cell lysate.

### 3.3. 2D vs. 3D EVs Have Different Effects on the Anti-Cancer Immunity Components

To evaluate possible differences between EVs released from 2D and 3D cell cultures (Appendix A), we analyzed the effect on innate immunity using both EVs from BT474 (luminal B) and HS578T (TNBC) cell lines as models. As reported in Figure 2 and Appendix A, no significant differences in the percentages of CD4+, CD8+ cells, NK cells (CD56+), NK-T cells (CD56+/CD3+) and monocytes (CD14+) were observed when PBMCs were stimulated with 3D BT474 (Figure 2A) and HS578T (Figure 3A) EVs. Interestingly, CD11b+/CD335+ NK cells were significantly less abundant in PBMCs stimulated with 3D BT474 EVs compared to both control and monolayer cells. Similarly, their degree of activation appeared to be significantly higher compared to both untreated cells and cells treated with EVs from BT474 2D, which confirmed the trend of the percentage of activated cells to total cells (Figure 2B). The greater effect of EVs arising from the spheroid formation of BT474 compared to the monolayer suggests a difference not in the composition of the EVs but in the concentration of the factors present in them, which may be more concentrated in EVs derived from spheroids. On the other hand, no significant differences were observed in CD11b+/CD335+ NK cells treated with EVs from both BT474 2D and the control, confirmed by the ratios of activated cells to total cells (Figure 2B). Instead, we observed a significant reduction in both CD11b+/CD335+ NK cells when stimulating PBMCs with EVs from both HS578T monolayer and spheroid models. Moreover, no differences were detected in cells treated with EVs from HS578T 2D and 3D (Figure 3A) compared to the percentage of the same cells activated in each condition. Surprisingly, regarding the ratios of activated cells to total cells, no significant differences were observed amongst PBMCs treated with EVs from HS578T 2D and 3D models compared to untreated cells (Figure 3B). Conversely, there was a significant difference in NK-T cells (CD56+/CD3+) in samples treated with EVs from the HS578T monolayer model compared to both treated and untreated cells with EVs from spheroids (Figure 3B).

### 3.4. EVs from Different Tumor Subtypes Have Similar Effects

The specific-subtype effects associated with EVs separated from both 2D and 3D models were evaluated using BT474 and HS578T cell lines. In monolayer models, our results showed a significant reduction in CD11b+/CD335+ NK cells in PBMCs treated with EVs separated from HS578T cells compared to BT474 cells (Figure 4A), while no significant differences were observed in the activation rate of the same cells. Moreover, a higher reduction in CD4+ T lymphocytes is evident in PBMCs treated with EVs from HS578T cells compared to EVs from BT474 cells. Furthermore, data showed significant differences in CD11b+/CD335+ NK and NK-T (CD56+/CD3+) cells treated with EVs from HS578T compared to BT474 monolayer cell lines (Figure 4B). On the contrary, no significant differences were observed in the percentage of CD11b+/CD335+ NK cells in PBMCs treated with EVs from both BT474 and HS578T spheroids (Figure 5A), or in the ratios of activated cells to total cells (Figure 5B).

### 3.5. EVs Modulate a Specific T-Reg Cell Regulation

To confirm the trend observed in the immune cells treated with EVs from breast cancer (2D and 3D models), we also evaluated the role of EVs on different subsets of T-regulatory cells. We aimed to further understand and analyze how T-reg cells behave when treated with EVs separated from BT474 and HS578T, both monolayer and spheroid. In Figure 6 and Appendix A, it is shown that no significant differences were observed in FoxP3+/Helios+, CD45RA+ and CD25+ T-regs. Interestingly, within the multiple T-reg population analyzed, the CD39+ were found to be significantly less abundant in samples treated with EVs separated from both monolayer cells and spheroids of BT474 compared to untreated samples. Conversely, there were no significant differences in CD39+ T-regs treated with EVs derived from HS578T.

## 4. Discussion

The tumor microenvironment (TME) is a puzzling scenario by which the tumor can grow and survive cancer therapies [5]. All the components that constitute the tumor microenvironment contribute to tumor survival in different ways. However, within the TME, the immune system is a complex ecosystem that contains several components, including adaptive and innate immune cells that have tumor-promoting and anti-tumor effects [29]. It is established that tumor cells can communicate with each other and with the immune system, represented by the TME, which facilitates the pro-tumorigenic phase. However, less is known regarding the role of extracellular vesicles in this scenario [30]. Indeed, EVs contain many signals and molecules that support the inactivation or hyperactivation of some components of innate immunity that accelerate tumor growth [31]. The anti-cancer immune response is the body’s first barrier against the onset of cancer and refers to the body’s natural defense mechanisms that help in the recognition and elimination of cancer cells [32]. Therefore, the immune system plays a crucial role in identifying and destroying cancer cells, preventing the development and progression of cancer. Various immune cells are involved in anti-cancer immunity, including [33] cytotoxic T cells (CD8+ T), which can directly recognize and destroy cancer cells by releasing cytotoxic molecules; B cells, which produce antibodies that can mark cancer cells for destruction, especially in antibody-dependent cellular cytotoxicity (ADCC); natural killer (NK) cells, which can directly kill cancer cells and help modulate the immune response; and dendritic cells, which help activate T cells by presenting cancer cell antigens to them.

Within those, NK cells have a key role in anti-cancer immunity as they are an active component of the innate immune system and produce cytokines like interferon-gamma (IFN-γ) and tumor necrosis factor-alpha (TNF-α). These cytokines have various effects, including stimulating other immune cells such as macrophages and T cells or reducing the anti-cancer immune response [34,35,36].

In this study, we evaluated the effect of EVs released from two breast cancer cell model systems, the BT474 (luminal B) and HS578T (TNBC) cell lines, on some of the components of anti-cancer immunity. Specifically, we focused on the effect of EVs released from 2D and 3D cell systems to further understand if a tumor-like cell organization would lead to a different effect than its bilayer counterpart.

Here, we demonstrated that CD335+/CD11b+ NK cells were significantly reduced when stimulating the PBMCs of healthy subjects with EVs derived from BT474 2D and 3D cell cultures, probably due to their significant hyperactivation in the presence of the EVs derived from BT474 spheroids. This difference might be correlated to a potential higher amount of pro-inflammatory, pro-tumorigenic factors and soluble molecules carried by EVs from spheroids, which can trigger the hyperactivation of CD335+/CD11b+ NK cells and consequently their reduction compared to EVs from the monolayer model. The greater effect of EVs arising from the spheroid formation of BT474 compared to the monolayer suggests a difference not in the composition of the EVs but in the concentration of the factors present in them, which may be more concentrated in EVs derived from spheroids. Obviously, future studies will focus on characterizing these factors in EVs by means of proteomic approaches. On the contrary, to the effect observed with BT474, in PBMCs treated with EVs released from HS578T cells, a significant reduction in the percentage of CD335+/CD11b+ NK cells was observed with separated EVs from both 2D and 3D cultures, which, however, was not associated with hyperactivation of the same cells under the conditions tested. The comparison of the two tumor subtypes showed significant differences in the percentage of CD335+/CD11b+ NK cells in PBMCs treated with EVs from HS578T cells, although these were lost when PBMCs were treated with EVs derived from 3D model. Based on these fundings, future studies are required to identify which proteins and/or nucleic acids encapsulated in EVs from 2D and 3D cell cultures are involved in TME organization. In our work, we have only used in vitro systems, but future studies will attempt to evaluate the effect of tumor-released EVs in in vivo systems, for example, by using organoids derived from breast cancer patients and characterizing the EVs released from them. In this way, an attempt will be made to translate the information obtained in this study into an environment closer to the real disease, such as organoids, to obtain personalized and more patient-oriented information.

Finally, the effect of EVs on T-reg cells was also evaluated. T-reg cells are a critical component of the immune system responsible for maintaining self-tolerance and preventing excessive immune reactions [37]. Moreover, T-regs can control several types of inflammatory responses by modulating the activity of different cells in the immune systems, which make them important targets for innovative diagnostic and therapeutic approaches in cancer [38]. Indeed, within the different subtypes, T-regs may have a pivotal role in shaping the immune landscape of the TME and aspects of anti-tumor immune responses [39].

Among T-reg cells, the CD39+ are unique, playing a vital role in suppressing excessive immune responses, which can be harmful to the body [40]. Specifically, the expression on the cellular surface of CD39 receptors allows them to convert ATP and ADP, often released during inflammation and tissue damage, into adenosine [41]. Furthermore, CD39+ T-reg cells can modulate the immune response by inhibiting the activation and function of other immune cells (e.g., T cells and dendritic cells) involved in the inflammatory response.

In this study, we observed that EVs derived from BT474 cells from 2D or 3D culture significantly reduce the amount of CD39+ T-reg cells. This effect might be in line with the results obtained on CD11b+/CD335+ NK cells, which, under the same conditions, were significantly more active. Again, the increased effect of EVs from BT474 spheroids compared to the monolayer counterpart could be due to a different concentration of factors acting on CD39+ T-reg activation. Therefore, this result could be correlated to a reduction in the inflammatory regulation of CD39+ T-reg cells. Furthermore, the greater effect of EVs derived from the BT474 spheroids compared to the BT474 2D model suggests that EV cargo may have different features from 3D cell culture. Overall, these preliminary data suggest only a potential explanation of the role that tumor-derived EVs may have on the immune system, which has not been published yet. However, future studies will focus on the characterization of EV factors using high-throughput analysis, such as proteomics.

## 5. Conclusions

Our results emphasize how important it is, in the study of the tumor microenvironment, to accurately evaluate one system, which could avoid potential bias because TME is a complex and dynamic ecosystem that surrounds and interacts with cancer cells within a tumor. We believe that our study may be a turning point for shedding a light on the effect of TME on the anti-cancer immune component to further develop personalized approaches in cancer diagnosis and treatment.

## Figures and Tables

**Figure 1 biology-12-01531-f001:**
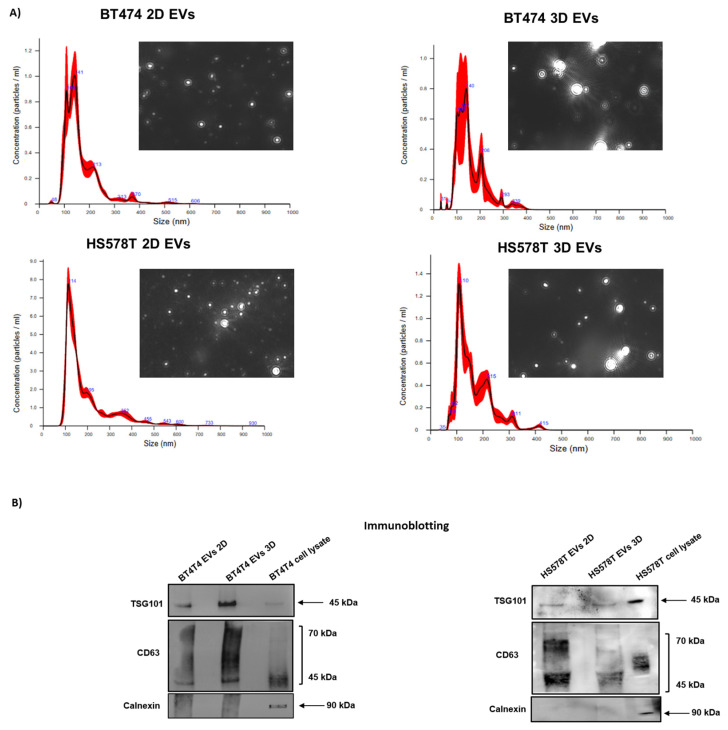
EV characterization from BT474 and HS578T 2D and 3D culture. (**A**) Representative graphs and pictures of particles in motion analyzed using Nanoparticle Tracking Analysis NS300 (NTA). (**B**) Immunoblotting analysis of EV markers in 2D and 3D breast cancer EVs from BT474 and HS578T. Equal amounts of protein (30 µg) were loaded for all the samples and analyzed for TSG101, CD81 and calnexin.

**Figure 2 biology-12-01531-f002:**
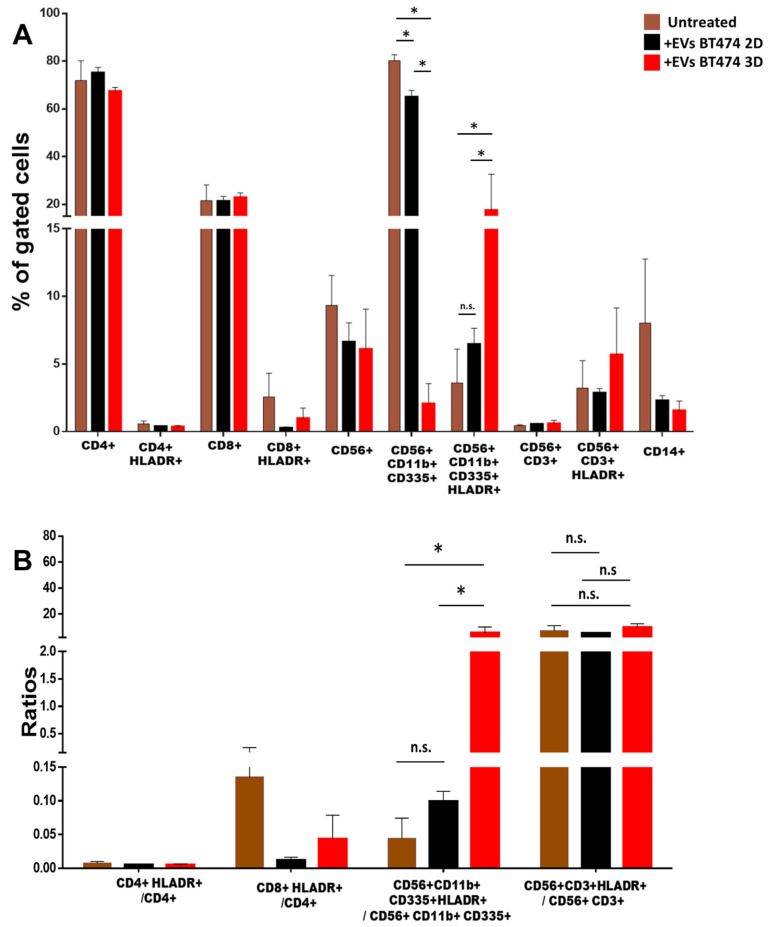
Effect of EVs from BT474 2D vs. 3D EVs on immune cells. (**A**) Histograms show the percentage of gated cells in PBMCs derived from 3 healthy volunteers treated with EVs derived from BT474 2D (black bars) and spheroids (red bars) with respect to the untreated PBMCs (light brown bars). (**B**) Ratio of activated cells with respect to the total cells. * = *p*-value = 0.05. n.s. = not significative.

**Figure 3 biology-12-01531-f003:**
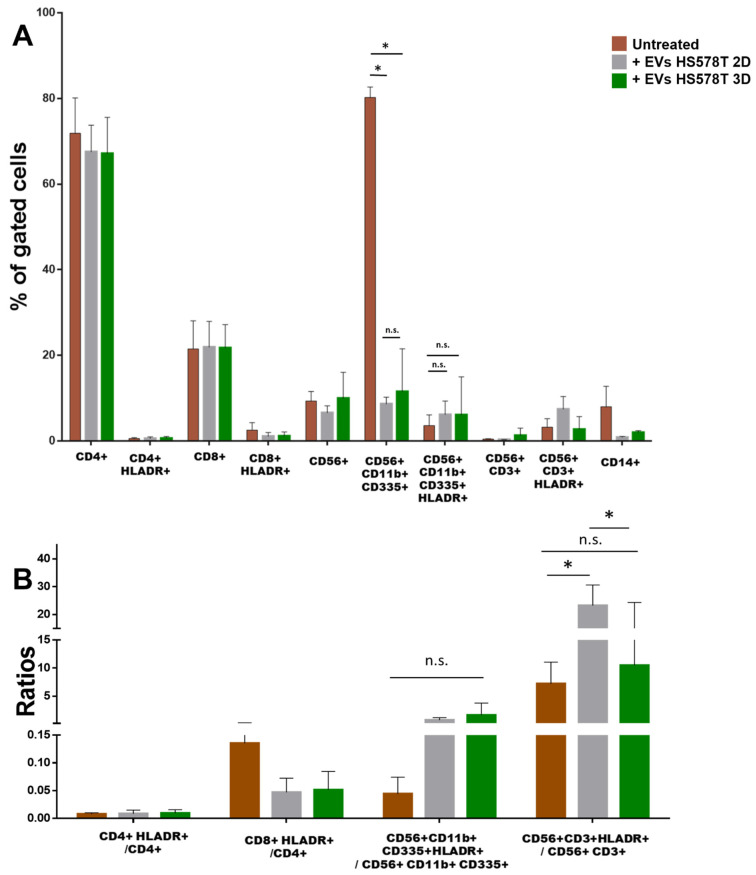
Effect of EVs from HS578T 2D vs. 3D on immune cells. (**A**) Histograms show the percentage of gated cells in PBMCs derived from 3 healthy volunteers treated with EVs derived from HS578T 2D (grey bars) and spheroids (green bars) with respect to the untreated PBMCs (light brown bars). (**B**) Ratio of activated cells with respect to the total cells. * = *p*-value = 0.05. n.s. = not significative.

**Figure 4 biology-12-01531-f004:**
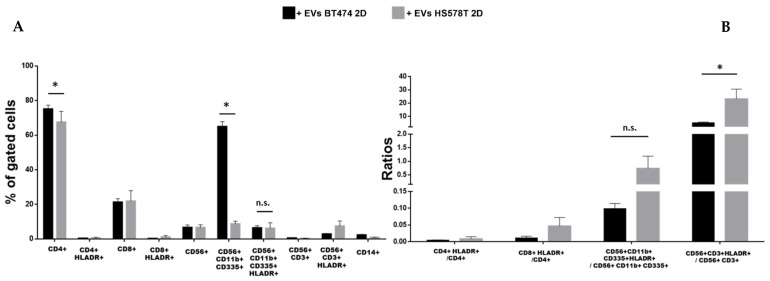
Effect of EVs from 2D breast cancer cells. (**A**) Histograms show the percentage of gated cells in PBMCs derived from 3 healthy volunteers treated with EVs derived from BT474 2D (black bars) and HS578T 2D (grey bars). (**B**) Ratio of activated cells with respect to the total cells. * = *p*-value = 0.05. n.s. = not significative.

**Figure 5 biology-12-01531-f005:**
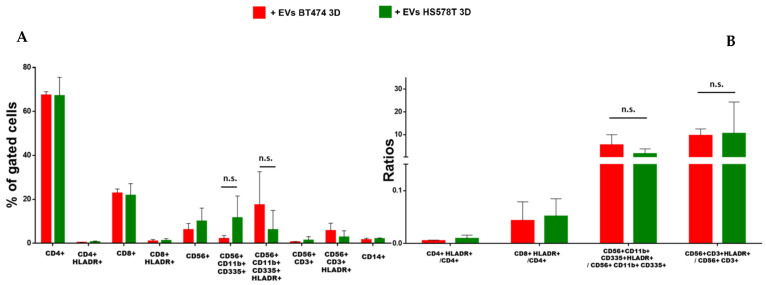
Effect of EVs from 3D breast cancer cells. (**A**) Histograms show the percentage of gated cells in PBMCs derived from 3 healthy volunteers treated with EVs derived from BT474 spheroids (red bars) and HS578T spheroids (green bars). (**B**) Ratio of activated cells with respect to the total cells. n.s. = not significative.

**Figure 6 biology-12-01531-f006:**
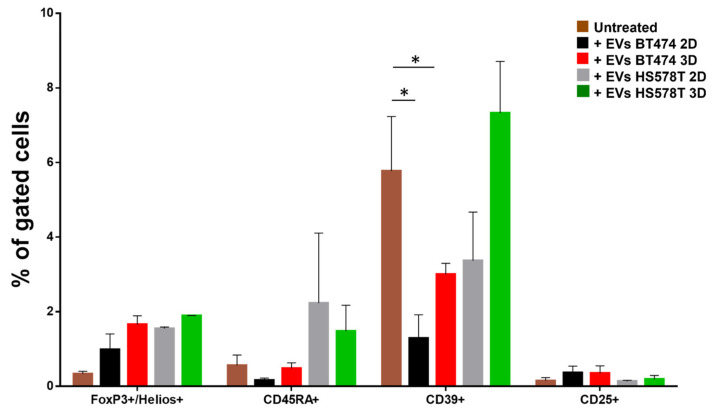
Representation of T-reg cells regulated by breast cancer EVs. Histograms show the percentage of gated T-reg cells in PBMCs derived from 3 healthy volunteers treated with EVs derived from BT474 2D (black bars), HS578T 2D (grey bars), BT474 spheroids (red bars) and HS578T spheroids (green bars) with respect to the untreated ones (brown bars). * = *p*-value = 0.05.

**Table 1 biology-12-01531-t001:** Particle concentration and size analyzed using NTA. Particle sizes and concentrations from BT474, HS578T 2D and 3D culture NTA analysis involved a minimum of five 60 s videos recorded for each sample. Three replicates of each sample were analyzed independently using NTA; data are presented as mean ± SEM.

Nanoparticle Tracking Analysis			
Particle Concentration (Particles/mL)		Particle Size	
BT474 2D	8.7 × 10^9^ ± 7.6 × 10^8^	BT474 2D	166.3 nm ± 72.9 nm
BT474 3D	6.9 × 10^10^ ± 2.3 × 10^10^	BT474 3D	156.2 nm ± 61.1 nm
HS578T 2D	6.3 × 10^10^ ± 3.2 × 10^9^	HS578T 2D	194.8 nm ± 112.8 nm
HS578T 3D	9.9 × 10^9^ ± 1.4 × 10^9^	HS578T 3D	163.5 nm ± 65.9 nm

## Data Availability

For the raw data utilized in this study, please contact the corresponding author L.C. (luigi.coppola@synlab.it).

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
