# Peer review of "Influence of Breast Cancer Extracellular Vesicles on Immune Cell Activation: A Pilot Study"

_biology, 2023, doi:10.3390/biology12121531_

Round 1
Reviewer 1 Report
Comments and Suggestions for Authors
In this manuscript, Santoro et al. described the effect of EVs from 2D and 3D culture of breast cancer cells on activation and presence of NK and T cells from PBMCs. The comparison between 2D and 3D would be novel point of view, however, there several points to be revised.
Major comments;
1. It is not clear whether EVs from 2D and 3D culture of the same cells had different effects. In Figure 2A, and B, the graph showed NK cells with HLADR was increased than NK cells without HLADR. The results look like EVs from 3D culture of BT474 has stronger effects for activation of NK cells, but the authors did not mention about it. The authors should described about it.
2. Similarly, in Figure 6, EVs from 3D culture seems to be less effective for reduction of CD39+ cells. However, the authors did not mention how 2D and 3D is different. The authors should described this point in text.
3. In discussion, the authors should discuss what is the difference between 2D and 3D, whether there is some difference, or no difference, and what would be the reason to explain the phenomena.
Minor comments;
Figures,
1. In validation of EVs, there are difference between NTA and SEM, the concentration and the size. The authors should explain about it.
2. In Figure 1A, the numbers of the graph on X- and Y- axis are so small and cannot see them. Please enlarge the characters.
3. In Figure 1B, it is not clear which culture the cell lysates came from, 2D ro 3D.
Comments on the Quality of English Language1. In introduction, the authors should explain what kind of breast cancer BT474 is, like triple positive while HS578T is triple negative, etc.
2. In materials and methods, immunoblotting analysis of EVs, the concentration of Triton and MgCl2 is not clear. Also subscript and superscript are not clear. 109 EVs, 105 PBMCs, etc, as well as the unit is not clear, 109 EVs /ml or not. The authors should clearly write these points.
Author Response
Reviewer 1
Open Review
(x) I would not like to sign my review report
( ) I would like to sign my review report
Quality of English Language
( ) I am not qualified to assess the quality of English in this paper
( ) English very difficult to understand/incomprehensible
( ) Extensive editing of English language required
( ) Moderate editing of English language required
(x) Minor editing of English language required
( ) English language fine. No issues detected
Yes |
Can be improved |
Must be improved |
Not applicable |
|
Does the introduction provide sufficient background and include all relevant references? |
( ) |
(x) |
( ) |
( ) |
Are all the cited references relevant to the research? |
( ) |
(x) |
( ) |
( ) |
Is the research design appropriate? |
(x) |
( ) |
( ) |
( ) |
Are the methods adequately described? |
( ) |
(x) |
( ) |
( ) |
Are the results clearly presented? |
( ) |
(x) |
( ) |
( ) |
Are the conclusions supported by the results? |
( ) |
(x) |
( ) |
( ) |
Comments and Suggestions for Authors
In this manuscript, Santoro et al. described the effect of EVs from 2D and 3D culture of breast cancer cells on activation and presence of NK and T cells from PBMCs. The comparison between 2D and 3D would be novel point of view, however, there several points to be revised.
Major comments;
- It is not clear whether EVs from 2D and 3D culture of the same cells had different effects. In Figure 2A, and B, the graph showed NK cells with HLADR was increased than NK cells without HLADR. The results look like EVs from 3D culture of BT474 has stronger effects for activation of NK cells, but the authors did not mention about it. The authors should described about it.
- Similarly, in Figure 6, EVs from 3D culture seems to be less effective for reduction of CD39+ cells. However, the authors did not mention how 2D and 3D is different. The authors should described this point in text.
- In discussion, the authors should discuss what is the difference between 2D and 3D, whether there is some difference, or no difference, and what would be the reason to explain the phenomena.
We would like to thank you the reviewer for the interesting suggestion. We agree with the reviewer's comments and have therefore added several sentences in the discussion section to better clarify the effects due to EVs from spheroids or monolayers of BT474 cells. Please check the sentences in red in the manuscript. However, we have not changed the results section since in this section we only explain the results without specific deductions which we were instead widely described in the discussion section.
Minor comments;
Figures,
- In validation of EVs, there are difference between NTA and SEM, the concentration and the size. The authors should explain about it.
We thank the reviewer for this comment. We have now rearranged the NTA table in the manuscript, therefore we hope it is now clarified for the readers.
- In Figure 1A, the numbers of the graph on X- and Y- axis are so small and cannot see them. Please enlarge the characters.
We thank the reviewer for the comment. We have now increased the size of the graphs in Figure 1A, as suggested.
- In Figure 1B, it is not clear which culture the cell lysates came from, 2D ro 3D.
We thank the reviewer for the comment. We used 2D cell lysate for each cell line as positive control and they are reported in red in material and methods section.
Comments on the Quality of English Language
- In introduction, the authors should explain what kind of breast cancer BT474 is, like triple positive while HS578T is triple negative, etc.
We thank the reviewer for the comment. We reported the cancer subtypes in red in the introduction section.
- In materials and methods, immunoblotting analysis of EVs, the concentration of Triton and MgCl2 is not clear. Also subscript and superscript are not clear. 109 EVs, 105 PBMCs, etc, as well as the unit is not clear, 109EVs /ml or not. The authors should clearly write these points.
We thank the reviewer for the comment. We have now modified the concentration of JS buffer components as final concentration used to lyse EVs. We also corrected the superscript and subscript in the manuscript. Moreover, regarding the unit used in section “PBMCs treatment with breast cancer EVs” , we referred to integer of EVs (109) as well as for PBMCs (105)
Reviewer 2 Report
Comments and Suggestions for Authors
The authors used two breast cancer cell lines derived EVs to induce PBMCs differentiation. They found EVs alter the presence of NK cell and Treg cells. However, all treatments had been done in vitro. In vivo experiments are missing to get the conclusion.
Major comments:
1. the flow cytometry results showed in supp. Fig 2 and 3: the positive and negative controls need to be added. And the gating process needs to be clearly labeled and described.
2. all the figure and table titles are missing.
3. Figure 1A, what is the Y-axis?
4. Fig. 1B, add the cell lysates panel. is the difference between 2D and 3D cultured EVs caused by cell proliferation alteration (culture condition).
Comments on the Quality of English Language
fine
Author Response
Reviewer 2
Open Review
(x) I would not like to sign my review report
( ) I would like to sign my review report
Quality of English Language
( ) I am not qualified to assess the quality of English in this paper
( ) English very difficult to understand/incomprehensible
( ) Extensive editing of English language required
( ) Moderate editing of English language required
(x) Minor editing of English language required
( ) English language fine. No issues detected
Yes |
Can be improved |
Must be improved |
Not applicable |
|
Does the introduction provide sufficient background and include all relevant references? |
( ) |
( ) |
(x) |
( ) |
Are all the cited references relevant to the research? |
( ) |
( ) |
(x) |
( ) |
Is the research design appropriate? |
( ) |
( ) |
(x) |
( ) |
Are the methods adequately described? |
( ) |
(x) |
( ) |
( ) |
Are the results clearly presented? |
( ) |
( ) |
(x) |
( ) |
Are the conclusions supported by the results? |
( ) |
( ) |
(x) |
( ) |
Comments and Suggestions for Authors
The authors used two breast cancer cell lines derived EVs to induce PBMCs differentiation. They found EVs alter the presence of NK cell and Treg cells. However, all treatments had been done in vitro. In vivo experiments are missing to get the conclusion.
Response: Thank you the reviewer for pointed out this aspect. The aim of our pilot study is to try to determine the role of tumour-released EVs in regulating the immune response and to emphasise the importance of the cell model to be used (2D vs 3D models). Future studies will focus on the use of PBMC derived from breast cancer patients in combination with the use of patient organoids. This, even more than mouse model systems, will shed light on the complex system regulating the tumour microenvironment, paving the way for new personalised and patient oriented diagnostic approaches. Some sentences were added to the discussion session to emphasize these limitations of our study.
Major comments:
- the flow cytometry results showed in supp. Fig 2 and 3: the positive and negative controls need to be added. And the gating process needs to be clearly labeled and described.
Response: We added the gating strategy process in the Figure legend of the Supplementary Figure 3 of the revised version of the manuscript. Negative control was added (see supplementary figure 4 of the revised version of the manuscript)
- all the figure and table titles are missing.
Response: We added the missing titles.
- Figure 1A, what is the Y-axis?
Response: We added the Y-axis label of Figure 1A.
- Fig. 1B, add the cell lysates panel. is the difference between 2D and 3D cultured EVs caused by cell proliferation alteration (culture condition).
Response: Thank you the reviewer for this consideration. Our aim was precisely to assess potential differences between the 2D and 3D models. It is likely that spheroids due to their morphological characteristics (much more tumour-like) may proliferate differently from their 2D counterparts. This could result in the release of similar EVs but probably with a higher concentration of hyper-activating molecules inside, which will be evaluated in future studies using proteomics approaches on EVs from the 2D and 3D models.
Reviewer 3 Report
Comments and Suggestions for Authors
The paper titled "Influence of Breast Cancer Extracellular Vesicles on Immune Cells Activation: A Pilot Study" is a significant contribution to the field of cancer biology. It explores the complex interactions within the tumor microenvironment, particularly focusing on the impact of breast cancer cell-derived extracellular vesicles on immune cell activation. The manuscript performed a detailed analysis of immune cell modulation, which could have implications for understanding cancer growth and treatment strategies. The findings suggest extracellular vesicles' potential role in altering immune system components, which could be crucial for developing personalized cancer treatments. I suggest the publication of this manuscript after a minor review:
General:
1- Contextual Background: The introduction could benefit from a more detailed explanation of the significance of extracellular vesicles in the context of breast cancer and the immune system. Adding information on EVs' known functions and mechanisms in cancer biology would provide a stronger foundation.
2- The study's objectives could be stated more explicitly. Clearly articulating the specific aims or hypotheses at the end of the introduction would help in setting clear expectations for the reader.
3- Methodological Rationale: A brief mention of why the chosen cell lines and methods (2D and 3D culture models) are particularly suited for this study could strengthen the introduction.
Results:
1- Page 7, line 2: The following text is in other language than English and needs to be translated “L’effetto Maggiore delle EVs derivanti dalla formazione sferoide delle BT474 rispetto al monolayer suggerisce una differenza non nella composizione delle EVs, bensì nella concentrazione dei fattori in esse presenti che potrebbero essere più concentrati nelle EVs derivanti da sferoidi.”
2- Did the author perform any statistical comparative analysis to demonstrate the differences between 2D and 3D cultures?
3- A significant reduction in CD11b+/CD335+ NK cells with HS578T EVs is mentioned, but how this compares across different conditions is unclear.
Discussion:
1- The discussion broadly categorizes immune responses without delving into the complexities and variations among different cancer types and individual patient responses. Cancer immunology can be highly variable, and a more nuanced approach to discussing immune responses would be beneficial.
2- The role of T-reg cells is discussed, but there seems to be an assumption about their behavior in the TME based on their general characteristics. A more cautious approach to interpreting their role in the context of cancer would be prudent, especially given the complex nature of immune regulation in cancer.
3- What are the specific functions and impacts of EVs within the TME?
Comments on the Quality of English LanguageN/A
Author Response
Reviewer 3
Open Review
(x) I would not like to sign my review report
( ) I would like to sign my review report
Quality of English Language
( ) I am not qualified to assess the quality of English in this paper
( ) English very difficult to understand/incomprehensible
( ) Extensive editing of English language required
( ) Moderate editing of English language required
(x) Minor editing of English language required
( ) English language fine. No issues detected
Yes |
Can be improved |
Must be improved |
Not applicable |
|
Does the introduction provide sufficient background and include all relevant references? |
( ) |
(x) |
( ) |
( ) |
Are all the cited references relevant to the research? |
(x) |
( ) |
( ) |
( ) |
Is the research design appropriate? |
(x) |
( ) |
( ) |
( ) |
Are the methods adequately described? |
(x) |
( ) |
( ) |
( ) |
Are the results clearly presented? |
( ) |
(x) |
( ) |
( ) |
Are the conclusions supported by the results? |
(x) |
( ) |
( ) |
( ) |
Comments and Suggestions for Authors
The paper titled "Influence of Breast Cancer Extracellular Vesicles on Immune Cells Activation: A Pilot Study" is a significant contribution to the field of cancer biology. It explores the complex interactions within the tumor microenvironment, particularly focusing on the impact of breast cancer cell-derived extracellular vesicles on immune cell activation. The manuscript performed a detailed analysis of immune cell modulation, which could have implications for understanding cancer growth and treatment strategies. The findings suggest extracellular vesicles' potential role in altering immune system components, which could be crucial for developing personalized cancer treatments. I suggest the publication of this manuscript after a minor review:
Response: thank you the reviewer for the positive comments.
General:
- Contextual Background: The introduction could benefit from a more detailed explanation of the significance of extracellular vesicles in the context of breast cancer and the immune system. Adding information on EVs' known functions and mechanisms in cancer biology would provide a stronger foundation.
Response: We added some sentences and references to add more explanation on EVs function in cancer biology.
- The study's objectives could be stated more explicitly. Clearly articulating the specific aims or hypotheses at the end of the introduction would help in setting clear expectations for the reader.
Response: We clarified the objectives of the study at the end of the introduction section.
- Methodological Rationale: A brief mention of why the chosen cell lines and methods (2D and 3D culture models) are particularly suited for this study could strengthen the introduction.
Response: We added the motivation of cell line choice in the introduction section of the revised version of the manuscript.
Results:
- Page 7, line 2: The following text is in other language than English and needs to be translated “L’effetto Maggiore delle EVs derivanti dalla formazione sferoide delle BT474 rispetto al monolayer suggerisce una differenza non nella composizione delle EVs, bensì nella concentrazione dei fattori in esse presenti che potrebbero essere più concentrati nelle EVs derivanti da sferoidi.”
Response: Thank you the reviewer for pointed out this mistake. We apologize for this unmentionable error. We modified the text using English language.
- Did the author perform any statistical comparative analysis to demonstrate the differences between 2D and 3D cultures?
Response: We applied the one way ANOVA test to determine the statistical differences between the 2D and 3D EVs effects on healthy PBMC (Figure 2 and 3). No other comparative statistical analyses were carried out between the two models used.
- A significant reduction in CD11b+/CD335+ NK cells with HS578T EVs is mentioned, but how this compares across different conditions is unclear.
Response: Thank you the reviewer for this suggestion. We limited ourselves to observing the effect of EVs from HS578T on healthy PBMCs. There are no differences between the two proposed models (2D vs 3D) so we cannot make any considerations at present. Future studies will focus on characterising the factors present in EVs from HS578T using proteomics approaches to try and clarify the absence of differences between the two models used.
Discussion:
- The discussion broadly categorizes immune responses without delving into the complexities and variations among different cancer types and individual patient responses. Cancer immunology can be highly variable, and a more nuanced approach to discussing immune responses would be beneficial.
Response: we added some sentences in the discussion session to emphasize this aspect.
- The role of T-reg cells is discussed, but there seems to be an assumption about their behavior in the TME based on their general characteristics. A more cautious approach to interpreting their role in the context of cancer would be prudent, especially given the complex nature of immune regulation in cancer.
Response: thank you the reviewer for the valuable suggestion. We have reworded the paragraph on T-reg in the discussion section to make the considerations more cautious with regard to the involvement of T-reg cells in the regulation of carcinogenesis processes.
- What are the specific functions and impacts of EVs within the TME?
Response: We added some sentences in the discussion session of the revised version of the manuscript regarding the impact of EVs in the Tumor microenvironment.
Round 2
Reviewer 2 Report
Comments and Suggestions for Authors
The authors sufficiently addressed all the comments.